# A Structural Equation Modelling Approach to Determine Factors of Bone Mineral Density in Korean Women

**DOI:** 10.3390/ijerph182111658

**Published:** 2021-11-06

**Authors:** Minkyung Je, Hyerim Kim, Yoona Kim

**Affiliations:** 1Department of Food and Nutrition, Gyeongsang National University, Jinju 52828, Korea; alsrud4687@naver.com (M.J.); nayana2841@naver.com (H.K.); 2Department of Food and Nutrition, Institute of Agriculture and Life Science, Gyeongsang National University, Jinju 52828, Korea

**Keywords:** structural equation modelling, bone mineral density, nutrient intake, premenopausal women and postmenopausal women

## Abstract

Background: no studies have assessed the associations of nutrient intake, physical activity, age, and body mass index (BMI) with bone mineral density (BMD) using structural equation modelling (SEM) in Korean women. The aim of this study was to examine the effects of nutrient intakes, physical activity, and body mass index (BMI) on BMD in Korean premenopausal and postmenopausal women, with the SEM approach, based on the fourth and fifth Korea National Health and Nutrition Examination Surveys (KNHANES) 2008–2011. Methods: SEM analysis was performed with 4160 women (2863 premenopausal women and 1297 postmenopausal women) aged 30–75 years in order to investigate total, direct, or mediating effects of nutrient intake, physical activity, age, and BMI on BMD. Model sensitivity to external misspecification and statistical significance of SEM was determined by phantom variables and bootstrapping. Reliability assessment of the SEM was done by Cronbach’s alpha. Results: a direct effect of minerals (potassium, calcium, and phosphorus) on BMD (total femur, femoral neck, lumbar spine, and whole body) was observed in premenopausal and postmenopausal women (*p* = 0.045 and *p* = 0.048, respectively). Age and BMI showed a total effect on BMD in premenopausal and postmenopausal women (*p* = 0.002, respectively). Conclusions: our study suggests that mineral intake (potassium, calcium, and phosphorus), age, and BMI are major contributors to BMD in Korean premenopausal and postmenopausal women aged 30–75 years.

## 1. Introduction

The world population is rapidly ageing. The number of persons aged over 60 years was 962 million in 2017 worldwide. This number is expected to double to nearly 2.1 billion in 2050. In 2050, the persons aged over 60 years will account for 35 percent of the population in Europe, 28 percent in Northern America, 25 percent in Latin America [1,2]. The fastest ageing Organization for Economic Cooperation and Development (OECD) countries are Greece, Korea, Poland, Portugal, Slovenia, and Spain, while the fastest ageing non-OECD countries are Brazil, China, and Saudi Arabia [3].

Bone loss occurs as people get older. A reduced bone mineralization caused by disproportion of bone resorption and bone mineralization leads to osteopenia [4,5]. People with osteopenia have a higher risk of osteoporosis [4]. Osteoporosis is characterised by decreased bone mineral density (BMD) and bone strength [6]. People with osteoporosis are more likely to have fracture risk. Global hip fracture is projected to increase from 37 percent in 2025 to 45 percent in 2050 in Asia, as the aging population is rapidly increasing [7].

Fractures elevate mortality rates [8] and impose an enormous burden on individuals (e.g., early retirement), society (e.g., work impact), and the health care system (e.g., hospitalizations, medication, rehabilitation, etc.) [9,10,11]. In this aspect, osteoporosis prevention is crucial.

BMD, as the quantity of mineral per volume of bone, is a predictor of osteoporosis in humans [12]. BMD, in regard to the whole body, total hip (including femoral neck), and lumbar spine has been reported as predictors of osteoporosis [13,14].

Factors, including age, gender, gene, hormone, medication, smoking, physical activity, nutritional status, and alcohol are associated with bone health [15,16]. Lifestyle factors can affect 20–40% of peak bone mass in adults. Lifestyle factors, including nutrient intake and physical activity, can be modifiable [15].

Structural equation modelling (SEM) is a relatively powerful, multivariate technique that quantifies the associations and interactions between multiple variables. SEM is beneficial for simultaneously examining all related pathways or associations between exogenous and/or endogenous (e.g., mediators) variables [17].

No studies exist assessing the associations of nutrient intake, physical activity, age, and body mass index (BMI) with BMD using SEM in Korean women.

The aim of this study was to determine total, direct, and mediating effects of nutrient intake, physical activity, age, and BMI on BMD in Korean premenopausal and postmenopausal women, aged between 30 and 75 years, based on the Korean National Health and Nutrition Examination Survey (KNHANES) 2008–2011.

## 2. Materials and Methods

### 2.1. Materials

The 4th and 5th KNHANES (2008–2011) were used in this study. The KNHANES is a national, cross-sectional health examination and survey in the Korean general population [18]. This survey protocol used a stratified, multistage-clustered probability sampling method for a representative sample of the noninstitutionalised Korean population. Systematic sampling is annually conducted with a new and different sample, of about 10,000 persons aged 1 year and over. The Korea Disease Control and Prevention Agency (KCDC) in South Korea has performed the KNHANES since 1998. The KNHANES consists of a health interview, a nutrition survey, and a health examination [18]. The health interview consists of components, including housing characteristics, medical conditions, socioeconomic status (income level, education level, occupation status), health care utilization, activity limitation, quality of life and injury, smoking, alcohol use, physical activity, mental health (mean daily sleep time), oral health, weight control, safety, and reproductive health for women. Income level was categorised as high, middle–high, middle–low, or low. Education level was categorised as less than elementary school, middle school, high school, or more than university. Occupation status was divided into “yes” for the employed and “no” for the unemployed and economically inactive population, which was based on the question: “Have you worked for more than an hour for income or for more than 18 h as an unpaid family worker in the past week? You need to answer “yes” if you are temporarily on leave from your work”. The question on the mean daily sleep time was “How much hours per day do you usually sleep? You need to write the mean daily sleep time”. The question on smoking status was categorised into non-smoker, former smoker, and current smoker during one’s lifetime. Subjects were asked if they never smoked or if they smoked in the past (yes) or if they currently smoked (yes) during their lifetime. If subjects answered “yes”, they chose one of the following answers: less than 5 packs (100 cigarettes) or more than 5 packs (100 cigarettes). The question on alcohol intake was categorised into non-drinker (no) and drinker (yes) during the lifetime based on “Have you ever had more than one drink in your lifetime?”. The health examination consisted of components, including body measurements, blood pressure, laboratory test (blood and urine), dental measurement, vision, retinal photo and visual field, audiometry, spirometry, balance, bone density and body composition, chest, and knee and hip-joint X-ray. The nutrition survey consisted of components, including dietary behaviour, dietary supplement use, food security, food frequency, and food and dietary intake [18]. A detailed explanation on the KNHANES is available at http://knhanes.kdca.go.kr (accessed on 24 February 2021).

Data on overall health levels, health-related awareness and behaviours, and food and nutrition intake from a sample population were statistically analysed.

### 2.2. Participants

The participants included in this study were premenopausal and postmenopausal women aged over 30–75 years. The excluded participants were women aged <30 or ≥76 years (n = 7751) and men (n = 17,195). Moreover, we excluded missing data on BMD measurement (n = 4501), disease status (arthritis, pulmonary tuberculosis, asthma, renal failure, diabetes mellitus, thyroid dysfunction, cancer, liver cirrhosis, or hepatitis (type B and C)) (n = 3387), missing data on anthropometric parameters (height, weight, and waist circumference) (n = 18), missing data on nutrition intake survey (n = 367), or missing data on physical activities, income level, education level (n = 71), and oestrogen use and ovariectomy status (n = 303). Finally, 4160 subjects (2863 premenopausal women and 1297 postmenopausal women) were included in the statistical analysis, as shown in Figure 1.

This present study was performed in accordance with the guidelines of the Declaration of Helsinki. KNHANES IV–V was approved by the Research Ethics Review Committee of KCDC (2008-04EXP-01-C, 2009-01CON-03-2C, 2010-02CON-21-C; 2011-02CON-06-C).

The ethical review and approval were waived for this study. Written informed consent was obtained from all subjects involved in the Study.

### 2.3. Measurements of Anthropometric Parameters and Bone Mineral Density

For SEM analysis, we obtained data from KNHANES IV–V, including demographic characteristics (age, income level, education level, occupation status, mean daily sleep time, smoking status, alcohol intake), anthropometric characteristics (height, weight, waist circumference, and BMI), physical activities (high-intense physical activity, moderate-intense activity, and regular walking), BMD (total femur, femoral neck, lumbar spine and whole body), and nutrient intakes (energy, water, protein, carbohydrate, calcium, phosphorus, potassium, total vitamin A, retinol only, and vitamin C).

Height (cm) of participants was measured to the nearest 0.1 cm. Body weight (kg) was measured to the nearest 0.1 kg, with the participant wearing light clothing without shoes.

BMI was calculated as weight (kg)/height squared (m^2^). Waist circumference (WC) was measured according to the World Health Organization (WHO) guideline.

Participants were asked how frequently they exercised weekly for the physical activity assessment with Korean version of the international physical activity questionnaire (IPAQ), categorised into “yes” or “no”.

Subjects who exercised vigorously for >20 min, at least three times a week, or moderate exercise, or walking for >30 min, at least five times a week, were considered as doing regular exercise or “yes” subjects.

Participants who did vigorous exercise for >20 min, at least three times a week, were considered as doing high intense physical activity. Participants who did a moderate exercise for >30 min, at least five times a week, were considered as doing moderate intense physical activity. Walking was considered as walking for >30 min at least five times a week.

BMD (g/cm^2^) measurements at total femur, femoral neck, lumbar spine (L1-L4), and whole body were done by a dual energy X-ray absorptiometry (DXA, Discovery. QDR 45000; Hologic Inc., Waltham, MA, USA). The WHO T-score criteria for Asians were used for diagnosis of osteopenia (–2.5 < T-score < –1.0) and osteoporosis (T-score ≤ –2.5).

### 2.4. Nutrient Intakes

Trained dietitians administered a multiple-pass 24-h dietary recall questionnaire for nutrient intake assessment, including food, energy, water, protein, fat, carbohydrate, fibre, ash, calcium, phosphorus, iron, sodium, potassium, total vitamin A, carotene, retinol only, thiamine, riboflavin, niacin, and vitamin C. The food composition table from the Korean National Rural Development Institute was used to estimate nutrient intake [19].

This study used observational variables of nutrient intake (energy, water, protein, carbohydrate, calcium, phosphorus, potassium, total vitamin A, retinol only, and vitamin C) for SEM analysis, using Cronbach’s alpha, which is an advantageous method in order to increase reliability, considering the principle of parsimony.

### 2.5. Structural Equation Modelling

We determined the related pathway or relationships between independent and/or dependent (e.g., mediators) variables based on results obtained after a reliability and feasibility analysis, using SEM on the AMOS 27.0 (IBM, Chicago, IL, USA).

The total, direct, and indirect effects of the major variables on BMD variables were investigated after the structural model fit was determined. The statistical significance of the indirect effect was tested with the bootstrapping technique to assess the mediating effect. Phantom variables were used to determine the mediating effects in multiple mediator models.

The reliability of multi-item scale variables was assessed with Cronbach’s alpha correlation coefficient. The reliability was elevated by omitting an item with a high value among Cronbach’s alpha values.

The full information maximum likelihood method was used for the research model evaluation. Absolute fit indices (Χ^2^ (Chi-square), Χ^2^/df (relative Chi-square), root mean square error of approximation (RMSEA)), and incremental fit indices (Tucker–Lewis Index (TLI) and comparative fit index (CFI)) were used for the model fit examination.

The bootstrapping technique was applied for the significance.

A total of 1000 samples were used for bootstrapping. Percentile confidence intervals were set at 95%. Bias-corrected confidence intervals were set at 95%.

### 2.6. Statistical Analysis

Before performing SEM procedures, missing values were treated through data pre-processing process. The processing of missing data is typically performed by removing or replacing, but in this study, all analyses were performed by removing missing values.

The Kolmogorov–Smirnov test, Q–Q plots, and histograms were used to test for the normality distribution. Variables that were not normally distributed were log-transformed.

To examine variables of general characteristics, education level, income level, occupation status, average sleep duration, alcohol consumption, smoking status, physical activity, and bone health status, the chi-squared test was used for categorical variables. We used two sample *t*-tests when variables were normally distributed. We used the Mann–Whitney U test when variables were not normally distributed.

Partial correlation coefficient was used to clarify the association between nutritional intake (energy, carbohydrate, protein, water, calcium, phosphorus, potassium, total vitamin A, retinol only, and vitamin C) and BMD (total femur, femoral neck, lumbar spine, and whole body) at each site to control the variables (age, BMI, physical activities, and alcohol consumption). The statistical analysis was performed with SPSS 27.0 (IBM, Chicago, IL, USA).

## 3. Results

### 3.1. Participant Characteristics

As shown in Table 1, a total of 4160 subjects aged 30–75 years were analysed in this study. The number of premenopausal women aged 30–39 years was 1515 (52.9%). The number of postmenopausal women aged 50–59 years was 583 (44.9%). The BMI was significantly higher in postmenopausal women than in premenopausal women (postmenopausal women median 23.6, premenopausal women median 22.5, respectively, *p* < 0.001).

Most of the subjects answered that they did not do high intense (vigorous) physical activity (premenopausal women’s n = 2431 (84.9%), postmenopausal women n = 1128 (87.0%), respectively) or that they did not do moderate intense physical activity (premenopausal women’s n = 2475 (86.4%), postmenopausal women n = 1099 (84.7%), respectively). Walking for 30 min or more and 5 days a week or more comprised 38.5% in premenopausal women (n = 1103) and 44.9% in postmenopausal women (n = 583).

### 3.2. Nutrient Intakes

As shown in Table 1, the median energy intake of postmenopausal women (median: 1510.9 Kcal/day) was higher than those of premenopausal women (median: 1210.9 Kcal/day) (*p* < 0.001). The energy intakes of premenopausal and postmenopausal women were lower than the 2020 Korean energy requirement. For reference, the average estimated energy requirement for the 2020 dietary reference intakes for Koreans (KDRIs) [20] was 1900 Kcal/day for women aged 30–49 years, 1700 Kcal/day for women aged 50–64 years, 1600 Kcal/day for women aged 65–74 years, and 1500 Kcal/day for women aged over 75 years. The recommended carbohydrates intake for the 2020 KDRIs [20] was 130 g/day for Korean women aged 30–75 years. In this study, recommended intakes of carbohydrates were 274.0 g/day in premenopausal women and 284.0 g/day in postmenopausal women (*p* = 0.014). Premenopausal and postmenopausal women consumed in excess of the recommended carbohydrate amounts. In this study, recommended intakes of protein was 62.4 ± 0.54 g/day in premenopausal women and 47.4 g/day (median) in postmenopausal women (*p* < 0.001). These amounts were higher than the recommended protein intake for premenopausal women (50 g/day) aged 30–75 years. However, the protein intake for postmenopausal women was lower than the recommended protein intake (50 g/day for women aged 30–75 years).

Water intake refers to the water content in food. In this study, recommended intakes of water were 796.6 mL/day (median) in premenopausal women and 639.4 mL/day (median) in postmenopausal women (*p* < 0.001). The water intake amounts were lower than recommended water intake (1000 mL/day for women aged 30–49 years, 900 mL/day for women aged 50–74 years and 800 mL/day for women aged over 75 years).

In the 2020 KDRIs [21], retinol activity equivalents (RAEs) were used as the unit of total vitamin A, but in fourth and fifth KNHANES, the unit of the total vitamin A was used as the retinol equivalent (RE). Therefore, this study used the 2010 KDRIs for only the total vitamin A among nutrients. Premenopausal women 581.7 μgRE/day (median) and postmenopausal women 491.7 μgRE/day (median) were found to be lower than the recommended total vitamin A intake (650 μgRE/day for women aged 30–49 years and 600 μgRE/day for women aged 50–75 years) (*p* < 0.001). Vitamin C intakes were 82.0 mg/day (median) for premenopausal women and 77.3 mg/day (median) for postmenopausal women. The recommended vitamin C intake for the 2020 KDRIs was 100 mg/day for Korean women aged 30–75 years. In this study, vitamin C intakes were lower than recommended vitamin C intakes [21].

Calcium intakes were 473.3 ± 5.8 mg/day for premenopausal women and 424.6 ± 8.3 mg/day for postmenopausal women. The calcium intake was lower than recommended calcium intake (550 mg/day for women aged 30–49 years and 600 mg/day for women aged 50–75 years). Phosphorus intakes were found in premenopausal women (median: 1002.4 mg/day) and postmenopausal women (median 894.6 mg/day) (*p* < 0.001). These intakes were higher than the recommended phosphorus intakes for women (700 mg/day) aged 30–75 years. The sufficient potassium intakes were 2847.2 ± 25.7 mg/day in premenopausal women and 2389.5 mg/day (median) in postmenopausal women (*p* < 0.001). Sufficient potassium intake for the 2020 KDRIs was 3500 mg/day for Korean women aged 30–75 years [22]. In this study, potassium intake was lower than sufficient potassium intake (Table 1).

### 3.3. Bone Health Status

The proportions of postmenopausal women with osteopenia at the total femur, femoral neck, and lumbar spine were 29.8%, 57.5%, and 45.0%, respectively. Moreover, 1.5%, 14.0%, and 27.8% of postmenopausal women had osteoporosis at the total femur, femoral neck, and lumbar spine, respectively. The proportions of premenopausal women with osteoporosis at the total femur, femoral neck, and lumbar spine were only 0.0%, 0.7%, and 0.8%, respectively **(**Table 1**)**.

### 3.4. Goodness of Fit Structural Equation Models and Variable Weights

Table 2 presents variable selection with Cronbach’s alpha. Nineteen variables were analysed with the reliability coefficient of Cronbach’s alpha, and were 0.755 for premenopausal women and 0.749 for postmenopausal women.

Table 3 presents the model fit evaluation results for BMDs in premenopausal and postmenopausal women. The model fit of premenopausal women was X^2^ = 1193.239 (df = 110; *p* < 0.001), which indicated the statistical significance of the X^2^/df. The model fitted the data (TLI = 0.949; CFI = 0.967; RMSEA = 0.059). Moreover, the model fit of postmenopausal women was X^2^ = 1123.090 (df = 124; *p* < 0.001), which indicated the statistical significance of the X^2^/df. The model fitted the data (TLI = 0.924; CFI = 0.945; RMSEA = 0.079).

Table 4 presents the standard regression weights of independent variables. The effects between the variables were compared with the absolute value if a value ranged from −1 to 1.

In premenopausal women, the most influential degree of physical activities was in the order of high intense physical activity (estimate = 0.501), moderate intense activity (estimate = 0.483), and regular walking (estimate = 0.323).

The most influential degree of energy, carbohydrate, and protein (E.C.P) was in the order of energy (estimate = 1.099), carbohydrate (estimate = 0.795), and protein (estimate = 0.727). The most influential degree of minerals was in the order of phosphorus (estimate = 0.990), potassium (estimate = 0.807), and calcium (estimate = 0.732). The most influential degree of vitamins was in the order of vitamin C (estimate = 0.503), total vitamin A (estimate = 0.390), and retinol only (estimate = 0.258).

In postmenopausal women, the most influential degree of physical activities was in the order of moderate intense physical activity (estimate = 0.645), regular walking (estimate = 0.258), and high moderate intense activity (estimate = 0.218).

The most influential degree of energy, carbohydrates, and protein (E.C.P.) was in the order of protein (estimate = 0.955), energy (estimate = 0.865), and carbohydrate (estimate = 0.606). The most influential degree of minerals was in the order of potassium (estimate = 0.907), phosphorus (estimate = 0.787), and calcium (estimate = 0.563). The most influential degree of vitamins was in the order of vitamin C (estimate = 0.896), vitamin A (estimate = 0.490), and retinol (estimate = 0.088).

### 3.5. Analysis of Direct and Total Effects of Each Component in Premenopausal Women with Bone Mineral Density

Figure 2 presents results of the direct and total pathways of variables between each component and BMD in premenopausal women using a SEM. Table 5 presents a summary of the direct and total pathways among major variables in premenopausal women based on findings from Figure 2. A direct association between minerals (potassium, calcium, and phosphorus) and BMD (total femur, femoral neck, lumbar spine, and whole body) was observed (*p* = 0.045). The direct association between vitamins (total vitamin A, retinol only and vitamin C) and BMD was observed (*p* = 0.016). The direct association between E.C.P. and BMD was also observed (*p* = 0.013). Moreover, the direct association between water intake and BMD was observed (*p* = 0.013). There were total effect associations between age and BMD (*p* = 0.002). It was observed that age and BMD not only had a direct effect, but also a total effect (*p* < 0.001 and *p* = 0.002, respectively). Moreover, there was a total effect of BMI on BMD (*p* = 0.002). It was observed that BMI and BMD not only had a direct effect, but also a total effect (*p* < 0.001 and *p* = 0.002, respectively).

### 3.6. Analysis of Mediating Pathways among Component in Premenopausal Women

Figure 3 presents a structural equation model analysing the mediating effect of each component and BMD in premenopausal women. Moreover, Table 6 presents a summary of the mediating effect among major variables in premenopausal women based on findings from Figure 3.

As shown in Table 6, the mediating effect of vitamins (total vitamin A, retinol only and vitamin C) on the association between age and BMD was statistically significant (*p* = 0.023). A significant mediating effect of E.C.P. in the association between age and BMD was found (*p* = 0.006). The mediating effect of E.C.P. on the association between BMI and BMD was also statistically significant (*p* = 0.028).

### 3.7. Analysis of Direct and Total Effects of Each Component in Postmenopausal Women with Bone Mineral Density

Figure 4 presents the direct and total effects of variables between each component and BMD in postmenopausal women using a SEM. Moreover, Table 7 presents a summary of the direct and total pathways among major variables in postmenopausal women based on findings from Figure 4.

A direct association between minerals (potassium, calcium, and phosphorus) and BMD (total femur, femoral neck, lumbar spine, and whole body) was observed (*p* = 0.048). The direct association between BMI and BMD was observed (*p* < 0.001). Moreover, the direct association between age and BMD was observed (*p* < 0.001). On the other hand, the direct association between E.C.P. and BMD was not statistically significant (*p* = 0.166). The direct association between vitamins (total vitamin A, retinol only and vitamin C) and BMD was not statistically significant (*p* = 0.393).

Direct associations between age and minerals, between age and vitamins, between age and water, and between age and E.C.P. were observed (*p* < 0.001, respectively). There was a total effect association between age and BMD (*p* = 0.002). It was observed that age and BMD not only had a direct effect, but also a total effect (*p* < 0.001 and *p* = 0.002, respectively).

### 3.8. Analysis of Mediating Pathways among Each Component in Postmenopausal Women

Figure 5 presents the mediating effect of variables between each component and BMD in postmenopausal women using a SEM. Moreover, Table 8 presents a summary of the mediating pathways among major variables in postmenopausal women based on findings from Figure 5. There was a direct association between age and minerals (*p* < 0.001), and between minerals and BMD (*p* = 0.048). However, a significant mediating effect of minerals on the association between age and BMD was not observed (*p* = 0.065).

### 3.9. Analysis of Partial Correlation Coefficients between Bone Mineral Density and Nutrient Intake in Premenopausal Women

In the premenopausal women, minerals, vitamins, and E.C.P. had a direct effect on BMD. Moreover, the mediating effect of vitamins and E.C.P. on the association between age and BMD was statistically significant.

Table 9 presents partial correlation coefficients to clarify the correlation between nutrient intake and BMD at each site. Total femur BMD was positively associated with carbohydrate after adjustment for age, BMI, physical activity, and alcohol consumption. In addition, total femur BMD was positively associated with potassium after adjustment for age, BMI, physical activity and alcohol consumption. Femoral neck BMD was positively associated with carbohydrate and energy after adjustment for age, BMI, physical activity, and alcohol consumption. Femoral neck BMD was positively associated with minerals (potassium and calcium) after adjustment for age, BMI, physical activity, and alcohol consumption. Lumbar spine BMD was not statistically significant in E.C.P., minerals (potassium, calcium, and phosphorus), vitamins (total vitamin A, retinol only, and vitamin C) and water after adjustment for age. In addition, whole body BMD was not statistically significant in E.C.P., minerals (potassium, calcium, and phosphorus), vitamins (total vitamin A, retinol only, and vitamin C), and water after adjustment for age.

### 3.10. Analysis of Partial Correlation Coefficients between Bone Mineral Density and Nutrient Intake in Postmenopausal Women

In postmenopausal women, minerals had a direct effect on BMD in a SEM. Table 10 presents an analysis of partial correlation coefficients to clarify correlation between nutrient intake and BMD at each site in postmenopausal women.

Total femur BMD was positively associated with minerals (potassium and calcium) after adjustment for age, BMI, physical activity, and alcohol consumption. In addition, total femur BMD was positively associated with water for age, BMI, physical activity, and alcohol consumption. Femoral neck BMD was positively associated with minerals (potassium, calcium, and phosphorus) after adjustment for age, BMI, physical activity, and alcohol consumption. Each of intake of total vitamin A, vitamin C, and water was positively associated with femoral neck BMD. Lumbar spine BMD was positively associated with calcium after adjustment for age. Lumbar spine BMD was positively associated with water after adjustment for age. Whole body BMD was positively associated with calcium after adjustment for age. In addition, whole body BMD was positively associated with water after adjustment for age.

## 4. Discussion

This study examined the major determinants of BMD investigating total, direct, and mediating effects of nutrient intake, physical activity, age, and BMI, on BMD in Korean premenopausal and postmenopausal women, aged 30–75 years, using a SEM based on the fourth and fifth KNHANES 2008–2011.

This present study found that age, BMI, E.C.P., minerals (potassium, calcium, and phosphorus), vitamins (total vitamin A, retinol only, and vitamin C) and water had direct effects on BMD in premenopausal women. This study also found that energy, carbohydrates, and protein mediated in the relationship between age and BMD, and in relationship between BMI and BMD in premenopausal women.

Fung et al., 2017 [23], observed no association between higher protein and fracture risk in postmenopausal women [23], while Rapuri et al., 2003 [24] observed a positive association between 3-year dietary protein intake as a percentage of energy and BMD in postmenopausal women aged 65–77 with calcium intakes > 408 mg/day [24].

Consistent with our results, a cross-sectional study of 994 healthy premenopausal women aged 45–49 years showed an increase in lumbar spine BMD after the highest consumption of zinc, magnesium, potassium, vitamin C, and fibre compared with the lowest consumption [25]. Premenopausal women who consumed higher amounts of fruit and milk in early adulthood had higher BMD compared to premenopausal women who consumed lower amounts of fruit and milk [25]. Vitamins mediated in the relationship between age and BMD.

Vitamin C is known to promote collagen matrix formation and osteoblast activity and inhibit osteoclast activity [26,27]. A meta-analysis showed higher dietary vitamin C intake was associated with increased BMD at femoral neck and lumbar spine [28]. On the other hand, other studies showed dietary vitamin C intake was not associated with BMD at the femoral neck and lumbar spine [29,30,31].

In the present study, we found a direct effect of vitamins (total vitamin A, retinol only, and vitamin C) on BMD only in premenopausal women, not in postmenopausal women. Partially consistent with this finding, the Iowa Women’s Health Study and the Women’s Health Initiative Observational Study showed little or no association between vitamin A or retinol intake and fracture risk in postmenopausal women [32,33]. On the other hand, several studies showed that higher vitamin A or retinol intake was positively associated with lower BMD fracture [34,35,36]. A meta-analysis of cohort studies showed vitamin A or retinol intake appeared to reduce total fracture risk, whereas hip fracture risk was elevated with vitamin A or retinol intake, indicating different influences on total fracture and hip fracture [37].

We found that mineral intake showed a direct effect on BMD in postmenopausal women. Consistent with our finding, Kong et al., 2017 [38] reported the highest potassium consumption favourably affected BMD at lumbar, total hip, and femur neck compared with the lowest potassium consumption in 4052 Korean postmenopausal women [38]. Studies of in vitro showed that acidosis promoted osteoclast activity and bone resorption [39,40,41]. Dietary potassium can play a key role in acid-base balance as an alkaline source, leading to calcium loss prevention from bone [42,43,44]. Potassium is rich in vegetables and fruits, dairy products, and nuts. Liu et al., 2015 [45], reported 6.4% and 4.8% BMD improvement for the whole body and the femoral neck with elevated daily consumption of 100 g/1000 kcal fruits in Chinese women aged over 60 years, indicating a 14.4% reduction in fracture in women [45]. A meta-analysis of cohort studies and randomised controlled trials showed reduced fracture risk with elevated consumption of more than 1 serving/day of fruits and vegetables in adults aged over 50 years [46]. Moreover, a cohort prospective study of Europe and the USA in adults aged over 60 years showed that fruit and vegetable consumption ranging from three to five servings/day lowered hip fracture risk by 39% compared with less than one serving/day of fruits and vegetables [47].

In this study, water showed direct effects on BMD in premenopausal women. Consistently, Vannucci et al., 2018 [48], reviewed the impact of mineral water high in calcium on bone health. They suggested drinking water high in calcium could be a good way to improve calcium bioavailability, leading to beneficial influence on bone health [48].

This present study found BMI had direct effects on BMD in premenopausal women. A study showed weight loss did not affect bone loss in overweight premenopausal women at either 1 g or 1.8 g/d of calcium intake [49].

The present study of models indicated a high goodness of fit with a TLI of 0.949, a CFI of 0.967, and a RMSEA of 0.059 for premenopausal and postmenopausal women, with a TLI of 0.924, a CFI of 0.945, and a RMSEA of 0.079 for postmenopausal women. This stable and appropriate model fit strongly supported our research hypotheses that nutrient intake, physical activity, age, and BMI would exert total, direct, and mediating effects on BMD in Korean premenopausal and postmenopausal women aged between 30 and 75 years. In summary of our findings, a direct effect of minerals (potassium, calcium, and phosphorus) on BMD (total femur, femoral neck, lumbar spine, and whole body) was observed in premenopausal and postmenopausal women. A direct effect and a total effect of age and BMI on BMD was observed in premenopausal and postmenopausal women. In premenopausal women, vitamin intake (total vitamin A, retinol only, and vitamin C), water intake, and E.C.P. intake showed a direct effect on BMD. E.C.P. intake and vitamin intake exerted a mediating effect on the association between age and BMD. E.C.P. intake showed a mediating effect on the association between BMI and BMD. Our findings support the results of previous studies [25,38,48], which showed the favourable influence of minerals on bone health. Moreover, our findings indicate that age, BMI, and mineral intake, which are major factors, can interact with each other, leading to influence on bone health. The findings of this study can aid the development of nutrition education and lifestyle modification strategies for the prevention of osteoporosis and fractures in aging women.

This study has some strengths. Thus far, no other study exist that have examined the determinants of BMD, with a SEM approach, in Korean women. The advantage of SEM allows us to powerfully assess measurement errors compared to a regression analysis. Examination of latent (unobserved) variables are enabled through observed variables [17].

This present study was the first study to investigate the effects of factors (nutrients, physical activity, BMI, and age) on BMD using the SEM approach in Korean women. The large sample size of the KNHANES was enough to determine the contributors to BMD with the model’s goodness-of-fit. Determinants of BMD were specifically examined by dividing premenopausal and postmenopausal women.

The limitation needs to be acknowledged. This present study was analysed based on a cross-sectional study of the KNHANES, limiting cause–effect association. It was hard to clarify the extent to which determinants affected BMD. In this study, the observed variables of the SEM were connected with endogenous latent variables (e.g., BMD) and exogenous latent variables (e.g., physical activity, E.C.P., minerals (potassium, calcium, and phosphorus), and vitamins (total vitamin A, retinol only and vitamin C), water, age, and BMI), which is hard to fully identify the causation of each observed variables.

Bias for non-response or measurement or information was not ruled out because this study was based on a cross-sectional study. The 24-h dietary recall for nutrient intake assessment might not be representative of general dietary habits of an individual. Residual confounding could be a limitation. However, a SEM analysis considers the measurement error of the measured variable and the perturbation error of the latent variable, so that the reflected value can be analysed. In this aspect, a SEM has the advantage of analysing values, considering errors compared with a logistic regression analysis.

Since this study used cross-sectional survey data, there is a limit on explaining the relationship between BMD and variables. Longitudinal and intervention studies, including the variables covered in this study, should be conducted in the future.

## 5. Conclusions

In conclusion, the present study showed that age, BMI, mineral intake (potassium, calcium, and phosphorus) were major determinants of BMD in Korean premenopausal and postmenopausal women. This result may have critical implications for bone health in Korean women. Further studies are warranted to verify the results.

## Figures and Tables

**Figure 1 ijerph-18-11658-f001:**
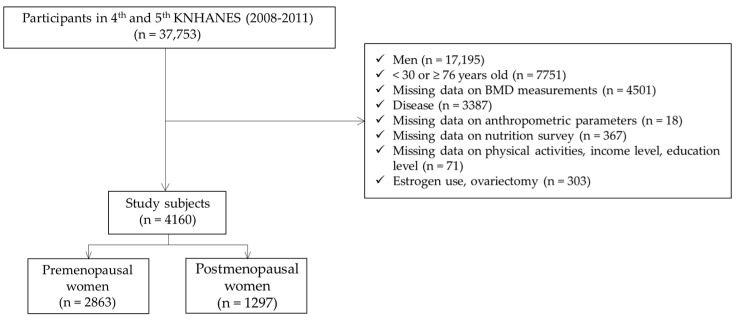
A flow diagram of the study’s inclusion and exclusion criteria of the subjects.

**Figure 2 ijerph-18-11658-f002:**
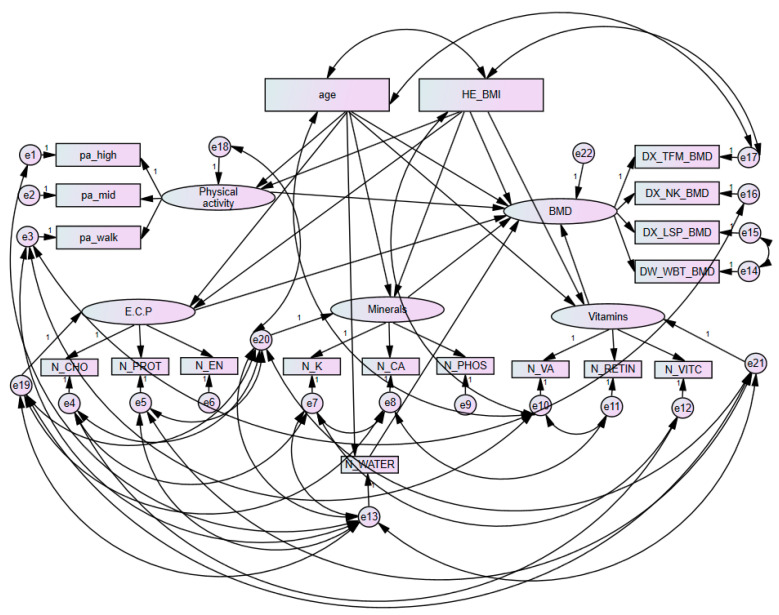
Results of direct pathways and total pathways with age, BMI, physical activity, and nutrient intakes in premenopausal women with bone mineral density. Errors are presented with e1, e2, e3, e4, e5, e6, e7, e8, e9, e10, e11, e12, e13, e14, e15, e16, e17, e18, e19, e20, e21 and e22. BMD, bone mineral density; N_CA, calcium; N_CHO, carbohydrate; N_EN, energy; E.C.P., energy, carbohydrates, and protein; DX_NK_BMD, femoral neck bone mineral density; pa_high, high intense physical activity; DX_LSP_BMD, lumbar spine bone mineral density; pa_mid, moderate intense physical activity; N_PHOS, phosphorus; N_K, potassium; N_PROT, protein; N_RETIN, retinol only; DX_TFM_BMD, total femur bone mineral density; N_VA, total vitamin A; N_VITC, vitamin C; N_WATER, water; pa_walk, regular walking; DX_WBT_BMD, whole body bone mineral density.

**Figure 3 ijerph-18-11658-f003:**
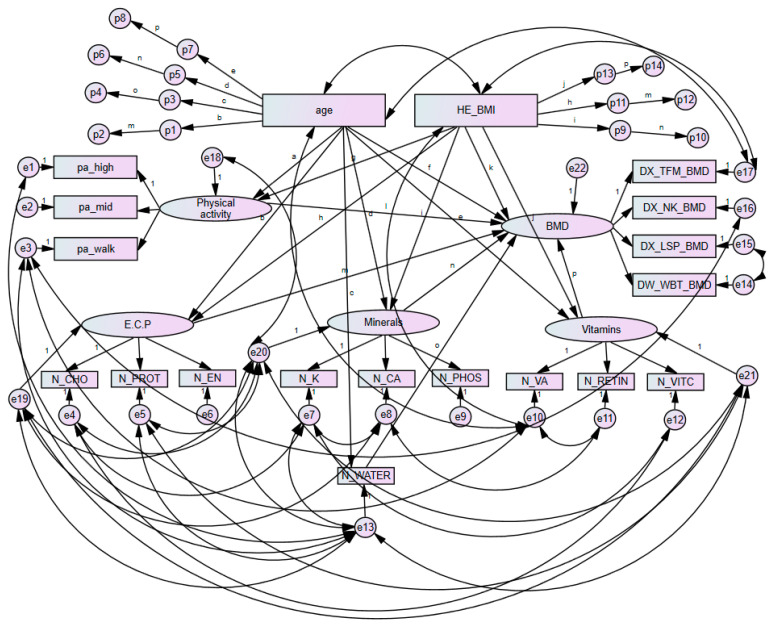
Results of mediated pathways with age, BMI, physical activity, and nutrient intakes in premenopausal women with bone mineral density. Errors are presented with e1, e2, e3, e4, e5, e6, e7, e8, e9, e10, e11, e12, e13, e14, e15, e16, e17, e18, e19, e20, e21 and e22. Phantom variables are presented with p1, p2, p3, p4, p5, p6, p7, p8, p9, p10, p11, p12, p13 and p14. BMD, bone mineral density; N_CA, calcium; N_CHO, carbohydrate; N_EN, energy; E.C.P., energy, carbohydrates, and protein; DX_NK_BMD, femoral neck bone mineral density; pa_high, high intense physical activity; DX_LSP_BMD, lumbar spine bone mineral density; pa_mid, moderate intense physical activity; N_PHOS, phosphorus; N_K, potassium; N_PROT, protein; N_RETIN, retinol only; DX_TFM_BMD, total femur bone mineral density; N_VA, total vitamin A; N_VITC, vitamin C; N_WATER, water; pa_walk, regular walking; DX_WBT_BMD, whole body bone mineral density.

**Figure 4 ijerph-18-11658-f004:**
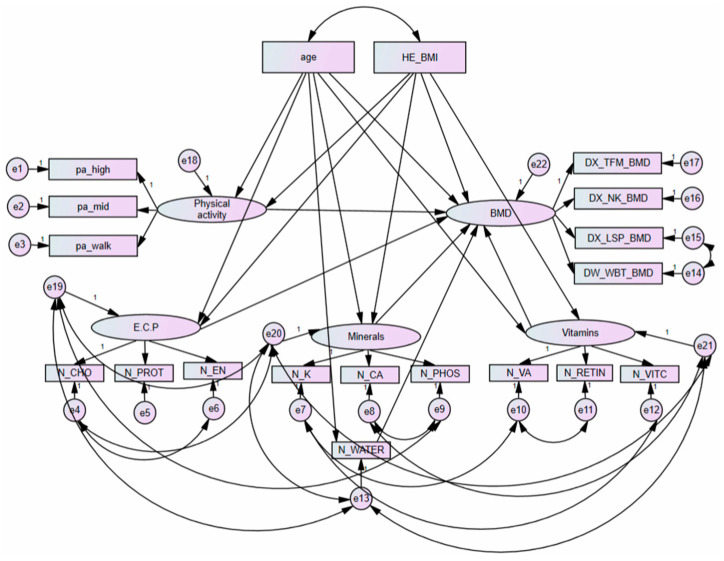
Results of direct pathways and total pathways with age, BMI, physical activity, and nutrient intakes in postmenopausal women with bone mineral density. Errors are presented with e1, e2, e3, e4, e5, e6, e7, e8, e9, e10, e11, e12, e13, e14, e15, e16, e17, e18, e19, e20, e21 and e22. BMD, bone mineral density; N_CA, calcium; N_CHO, carbohydrate; N_EN, energy; E.C.P., energy, carbohydrates, and protein; DX_NK_BMD, femoral neck bone mineral density; pa_high, high intense physical activity; DX_LSP_BMD, lumbar spine bone mineral density; pa_mid, moderate intense physical activity; N_PHOS, phosphorus; N_K, potassium; N_PROT, protein; N_RETIN, retinol only; DX_TFM_BMD, total femur bone mineral density; N_VA, total vitamin A; N_VITC, vitamin C; N_WATER, water; pa_walk, regular walking; DX_WBT_BMD, whole body bone mineral density.

**Figure 5 ijerph-18-11658-f005:**
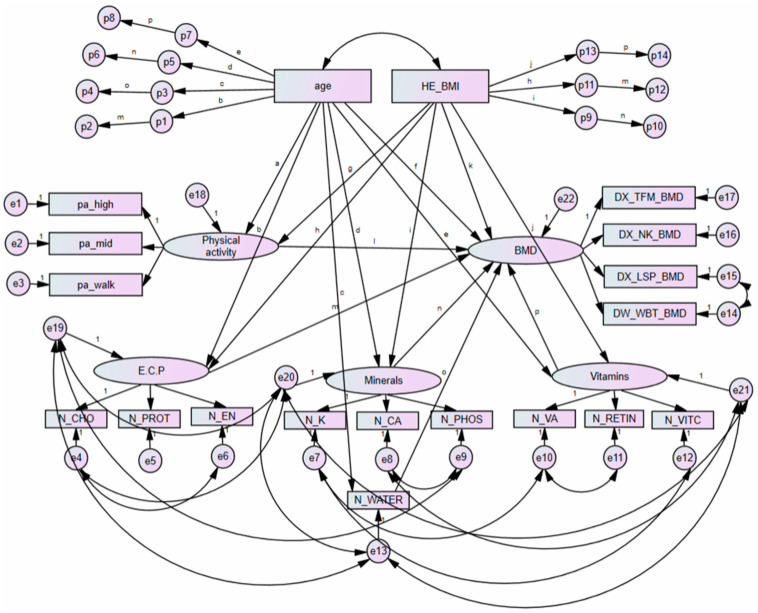
Results of mediated pathways with age, BMI, physical activity, and nutrient intakes in postmenopausal women with bone mineral density. Errors are presented with e1, e2, e3, e4, e5, e6, e7, e8, e9, e10, e11, e12, e13, e14, e15, e16, e17, e18, e19, e20, e21 and e22. Phantom variables are presented with p1, p2, p3, p4, p5, p6, p7, p8, p9, p10, p11, p12, p13 and p14. BMD, bone mineral density; N_CA, calcium; N_CHO, carbohydrate; N_EN, energy; E.C.P., energy, carbohydrates, and protein; DX_NK_BMD, femoral neck bone mineral density; pa_high, high intense physical activity; DX_LSP_BMD, lumbar spine bone mineral density; pa_mid, moderate intense physical activity; N_PHOS, phosphorus; N_K, potassium; N_PROT, protein; N_RETIN, retinol only; DX_TFM_BMD, total femur bone mineral density; N_VA, total vitamin A; N_VITC, vitamin C; N_WATER, water; pa_walk, regular walking; DX_WBT_BMD, whole body bone mineral density.

**Table 1 ijerph-18-11658-t001:** Baseline characteristics of the study subjects.

Variables	Women (n = 4160)		*p* Value
	Premenopausal (n = 2863) (%)	Postmenopausal (n = 1297) (%)	
Age (year)	39.0, 35.0	59.0, 53.0	<0.001
30–39	1515 (52.9)	6 (0.5)	
40–49	1175 (41.1)	85 (6.6)	
50–59	173 (6.0)	583 (44.9)	
60–69	0 (0.0)	437 (33.8)	
70–75	0 (0.0)	186 (14.4)	
Height (cm)	158.9 ± 5.4	154.0 ± 0.2	0.077
Weight (kg)	56.5, 51.6	55.8, 51.0	0.002
Body Mass Index (kg/m^2^)	22.5, 20.5	23.6, 21.8	<0.001
Waist circumference (cm)	75.1, 70.0	80.4, 75.2	<0.001
Education			<0.001
≤Elementary	120 (4.2)	730 (56.3)	
Middle school	212 (7.4)	217 (16.7)	
High school	1402 (49.0)	274 (21.1)	
≥University	1129 (39.4)	76 (5.9)	
Income level			0.635
Low	677 (23.6)	328 (25.3)	
Middle–low	728 (25.4)	324 (25.0)	
Middle–high	747 (26.1)	340 (26.2)	
High	711 (24.8)	305 (23.5)	
Occupation			0.047
Yes	1552 (54.2)	660 (50.9)	
No	1311 (45.8)	637 (49.1)	
Average sleep time per day			<0.001
<7 h	895 (31.3)	606 (46.7)	
7–9 h	1894 (66.2)	659 (50.8)	
>9 h	74 (2.6)	32 (2.5)	
Alcohol consumption status			<0.001
Yes	2598 (90.7)	890 (68.6)	
No	265(9.3)	407 (31.4)	
Smoking status			0.005
<100 cigarettes in lifetime	59 (2.1)	11 (0.8)	
≥100 cigarettes in lifetime	248 (8.7)	87 (6.7)	
Never smoked	2554 (89.2)	1198 (92.4)	
Don’t know	2 (0.01)	1 (0.1)	
Nutrient intakes (per day)			
Energy (Kcal)	1210.9, 872.4	1510.9, 1206.3	<0.001
Water (g)	796.6, 537.5	639.4, 409.3	<0.001
Protein (g)	57.51, 43.15	47.4, 35.1	<0.001
Carbohydrate (g)	274.0, 211.5	284.0, 223.0	0.014
Calcium (mg)	473.3 ± 5.8	424.6 ± 8.3	0.854
Phosphorus (mg)	1002.4, 766.0	894.6, 686.7	<0.001
Potassium (mg)	2607.22, 1916.15	2389.5, 1702.5	<0.001
Total vitamin A (μgRE)	581.7, 354.1	491.7, 244.2	<0.001
Retinol only (μg)	61.8, 18.8	20.4, 3.1	<0.001
Vitamin C (mg)	82.0, 49.5	77.3, 43.3	0.001
Physical activity			
High-intense			0.080
Yes	432 (15.1)	169 (13.0)	
No	2431 (84.9)	1128 (87.0)	
Moderate-intense			0.141
Yes	388 (13.6)	198 (15.3)	
No	2475 (86.4)	1099 (84.7)	
Regular walking			<0.001
Yes	1103 (38.5)	583 (44.9)	
No	1760 (61.5)	714 (55.1)	
Bone mineral density			
Total femur BMD (g/cm^2^)	0.903, 0.824	0.787, 0.717	<0.001
Femoral neck BMD (g/cm^2^)	0.760 ± 0.002	0.643 ± 0.003	0.740
Lumbar spine BMD (g/cm^2^)	0.995 ± 0.002	0.809 ± 0.004	<0.001
Whole body BMD (g/cm^2^)	1.143, 1.073	0.995, 0.922	<0.001
Bone health status			
Total femur			
Normal	2708 (94.6)	891 (68.7)	
Osteopenia	154 (5.4)	386 (29.8)	
Osteoporosis	1 (0.0)	20 (1.5)	
Femoral neck			
Normal	2055 (71.8)	370 (28.5)	
Osteopenia	788 (27.5)	746 (57.5)	
Osteoporosis	20 (0.7)	181 (14.0)	
Lumbar spine			
Normal	2316 (80.9)	352 (27.1)	
Osteopenia	525 (18.3)	584 (45.0)	
Osteoporosis	22 (0.8)	361 (27.8)	

Normally distributed values are presented as means ± standard errors. Non-normally distributed values are shown as medians and interquartile ranges. BMD, bone mineral density. Categorical variables were analysed by chi-squared test. Parametric variables were analysed by two samples *t*-test. Non-parametric variables were analysed by Mann–Whitney U test.

**Table 2 ijerph-18-11658-t002:** Variable selection with Cronbach’s α.

Cronbach’s Alpha If Variables Deleted			
Observed Variables	Variables	Premenopausal Women (n = 2863)	Postmenopausal Women (n = 1297)
Age	age	0.758	0.751
HE_BMI	Body mass index	0.758	0.751
Physical activity			
pa_high	High-intense	0.758	0.751
pa_mid	Moderate-intense	0.758	0.751
pa_walk	Regular walking	0.758	0.751
Nutrient intakes			
N_CHO	Carbohydrate	0.745	0.739
N_PROT	Protein	0.754	0.748
N_EN	Energy	0.696	0.695
N_K	Potassium	0.727	0.740
N_CA	Calcium	0.730	0.724
N_PHOS	Phosphorus	0.703	0.696
N_VA	Total vitamin A	0.730	0.716
N_RETIN	Retinol only	0.752	0.744
N_VITC	Vitamin C	0.749	0.741
N_WATER	Water	0.710	0.695
Bone mineral density			
DX_TFM_BMD	Total femur BMD	0.758	0.751
DX_NK_BMD	Femoral neck BMD	0.758	0.751
DX_LSP_BMD	Lumbar spine BMD	0.758	0.751
DX_WBT_BMD	Whole body BMD	0.758	0.751
Cronbach’s alpha		0.755	0.749

**Table 3 ijerph-18-11658-t003:** Evaluation of the model fit in study subjects with bone mineral density.

Subjects	Χ^2^	df	*p*	Χ^2^/df	TLI	CFI	RMSEA
Premenopausal	1193.239	110	<0.001	10.848	0.949	0.967	0.059
Postmenopausal	1123.090	124	<0.001	9.057	0.924	0.945	0.079

CFI, comparative fit index; df, degree of freedom; RMSEA, root mean square error of approximation; TLI, Tucker–Lewis index.

**Table 4 ijerph-18-11658-t004:** Standard regression weights of independent variables.

			Estimate	
	Code Name	Variables Name	Premenopausal Women	Postmenopausal Women
Physical activity	pa_high	High intense	0.501	0.218
	pa_mid	Moderate intense	0.483	0.645
	pa_walk	Regular walking	0.323	0.258
E.C.P	N_EN	Energy	1.099	0.865
	N_CHO	Carbohydrate	0.795	0.606
	N_PROT	Protein	0.727	0.955
Minerals	N_K	Potassium	0.807	0.907
	N_CA	Calcium	0.732	0.563
	N_PHOS	Phosphorus	0.990	0.787
Vitamins	N_VA	Total vitamin A	0.390	0.490
	N_RETIN	Retinol only	0.258	0.088
	N_VITC	Vitamin C	0.503	0.896

E.C.P., energy, carbohydrates, and protein.

**Table 5 ijerph-18-11658-t005:** Results of direct and total pathways in premenopausal women with bone mineral density.

	Pathways			β	S.E.	C.R.	*p*
Direct pathways	Age	→	Vitamins	−6.031	1.626	−3.708	<0.001
	HE_BMI	→	Physical activity	0.003	0.002	1.904	0.057
	HE_BMI	→	Minerals	−9.898	3.679	−2.690	0.007
	HE_BMI	→	Vitamins	−3.775	2.591	−1.457	0.145
	Age	→	Physical activity	0.003	0.001	3.463	<0.001
	HE_BMI	→	E.C.P	−0.701	0.310	−2.260	0.024
	Age	→	Minerals	−1.318	2.685	−0.491	0.623
	Age	→	E.C.P	−0.643	0.220	−2.919	0.004
	Age	→	N_WATER	−0.748	1.593	−0.469	0.639
	Physical activity	→	BMD	0.020	0.016	1.268	0.205
	Age	→	BMD	−0.002	0.000	−4.664	<0.001
	E.C.P	→	BMD	0.000	0.000	2.498	0.013
	Minerals	→	BMD	0.000	0.000	2.008	0.045
	Vitamins	→	BMD	0.000	0.000	−2.419	0.016
	HE_BMI	→	BMD	0.012	0.001	19.581	<0.001
	N_WATER	→	BMD	0.000	0.000	2.490	0.013
Total pathways	Age	→	BMD	−0.001	0.000	-	0.002
	HE_BMI	→	BMD	0.012	0.001	-	0.002

HE_BMI, body mass index; BMD, bone mineral density; C.R., critical ratio; E.C.P., energy, carbohydrates, and protein; β, standard regression coefficient; S.E., standard error; N_WATER, water.

**Table 6 ijerph-18-11658-t006:** Results of mediated pathways in premenopausal women with bone mineral density.

Mediated Pathways	B	*p*
Age → E.C.P. → BMD	0.000	0.006
Age → N_WATER → BMD	0.000	0.632
Age → Minerals → BMD	0.000	0.646
Age → Vitamins → BMD	0.001	0.023
HE_BMI → Minerals → BMD	0.000	0.072
HE_BMI → E.C.P. → BMD	0.000	0.028
HE_BMI → Vitamins → BMD	0.000	0.152

HE_BMI, body mass index; BMD, bone mineral density; E.C.P., energy, carbohydrates, and protein; B, regression coefficient; N_WATER, water.

**Table 7 ijerph-18-11658-t007:** Results of direct and total pathways in postmenopausal women with bone mineral density.

	Pathways			β	S.E.	C.R.	*p*
Direct pathways	Age	→	Vitamins	−10.931	1.495	−7.313	<0.001
	Age	→	N_WATER	−14.864	1.581	−9.405	<0.001
	Age	→	E.C.P	−1.705	0.191	−8.905	<0.001
	Age	→	Physical activity	−0.001	0.000	−1.281	0.200
	Age	→	Minerals	−38.321	4.330	−8.850	<0.001
	HE_BMI	→	Minerals	−4.638	7.645	−0.607	0.544
	HE_BMI	→	E.C.P	0.006	0.324	0.020	0.984
	HE_BMI	→	Physical activity	0.001	0.001	0.576	0.565
	HE_BMI	→	Vitamins	−0.002	2.700	−0.001	0.999
	N_WATER	→	BMD	0.000	0.000	−0.034	0.973
	Age	→	BMD	−0.007	0.000	−21.204	<0.001
	Physical activity	→	BMD	0.058	0.051	1.133	0.257
	E.C.P	→	BMD	0.000	0.000	−1.386	0.166
	Minerals	→	BMD	0.000	0.000	1.978	0.048
	Vitamins	→	BMD	0.000	0.000	0.855	0.393
	HE_BMI	→	BMD	0.011	0.001	12.928	<0.001
Total pathways	Age	→	BMD	−0.007	0.000	-	0.002
	HE_BMI	→	BMD	0.011	0.001	-	0.002

HE_BMI, body mass index; BMD, bone mineral density; C.R., critical ratio; E.C.P., energy, carbohydrates, and protein; β, standard regression coefficient; S.E., standard error; N_WATER, water.

**Table 8 ijerph-18-11658-t008:** Results of mediated pathways in postmenopausal women with bone mineral density.

Mediated Pathways	B	*p*
Age → E.C.P. → BMD	0.000	0.240
Age → N_WATER → BMD	0.000	0.991
Age → Minerals → BMD	0.000	0.065
Age → Vitamins → BMD	0.000	0.427
HE_BMI → Minerals → BMD	0.000	0.554
HE_BMI → E.C.P. → BMD	0.000	0.987
HE_BMI → Vitamins → BMD	0.000	0.969

HE_BMI, body mass index; BMD, bone mineral density; E.C.P., energy, carbohydrates, and protein; B, regression coefficient; N_WATER, water.

**Table 9 ijerph-18-11658-t009:** Partial correlation coefficients between nutrient intake and bone mineral density in premenopausal women.

BMD		Unadjusted	r	*p*		Adjusted	r	*p*
Total femur ^a^	E.C.P	Energy ^‡^	−0.003	0.885	E.C.P	Energy ^‡^	0.029	0.121
		Carbohydrate ^‡^	0.031	0.097		Carbohydrate ^‡^	0.046	0.014
		Protein ^†^	−0.029	0.122		Protein ^†^	0.003	0.872
	Minerals	Potassium ^†^	0.012	0.522	Minerals	Potassium ^†^	0.042	0.024
		Calcium ^†^	−0.001	0.947		Calcium ^†^	0.020	0.286
		Phosphorus ^‡^	−0.022	0.241		Phosphorus ^‡^	0.009	0.621
	Vitamins	Total vitamin A ^‡^	0.020	0.284	Vitamins	Total vitamin A ^‡^	0.016	0.392
		Retinol only ^‡^	−0.057	0.002		Retinol only ^‡^	−0.009	0.649
		Vitamin C ^‡^	0.028	0.133		Vitamin C ^‡^	0.032	0.085
	Water	Water ^‡^	0.004	0.840	Water	Water ^‡^	0.015	0.410
Femoral neck ^a^	E.C.P	Energy ^‡^	0.015	0.431	E.C.P	Energy ^‡^	0.043	0.021
		Carbohydrate ^‡^	0.037	0.047		Carbohydrate ^‡^	0.056	0.003
		Protein ^†^	−0.012	0.515		Protein ^†^	0.013	0.480
	Minerals	Potassium ^†^	0.017	0.370	Minerals	Potassium ^†^	0.046	0.014
		Calcium ^†^	0.017	0.359		Calcium ^†^	0.039	0.038
		Phosphorus ^‡^	−0.004	0.849		Phosphorus ^‡^	0.024	0.206
	Vitamins	Total vitamin A ^‡^	0.026	0.164	Vitamins	Total vitamin A ^‡^	0.018	0.337
		Retinol only ^‡^	−0.023	0.221		Retinol only ^‡^	−0.011	0.548
		Vitamin C ^‡^	0.020	0.293		Vitamin C ^‡^	0.030	0.103
	Water	Water ^‡^	0.017	0.352	Water	Water ^‡^	0.029	0.120
Lumbar spine ^b^	E.C.P	Energy ^‡^	−0.004	0.834	E.C.P	Energy ^‡^	−0.003	0.864
		Carbohydrate ^‡^	0.020	0.292		Carbohydrate ^‡^	0.011	0.571
		Protein ^†^	−0.015	0.423		Protein ^†^	−0.015	0.420
	Minerals	Potassium ^†^	0.000	0.994	Minerals	Potassium ^†^	0.000	0.987
		Calcium ^†^	−0.006	0.743		Calcium ^†^	−0.006	0.747
		Phosphorus ^‡^	−0.029	0.123		Phosphorus ^‡^	−0.030	0.113
	Vitamins	Total vitamin A ^‡^	0.021	0.254	Vitamins	Total vitamin A ^‡^	0.017	0.371
		Retinol only ^‡^	−0.031	0.092		Retinol only ^‡^	−0.010	0.592
		Vitamin C ^‡^	0.011	0.547		Vitamin C ^‡^	−0.024	0.204
	Water	Water ^‡^	0.014	0.464	Water	Water ^‡^	−0.003	0.868
Whole body ^b^	E.C.P	Energy ^‡^	−0.025	0.185	E.C.P	Energy ^‡^	−0.014	0.444
		Carbohydrate ^‡^	−0.011	0.568		Carbohydrate ^‡^	−0.011	0.563
		Protein ^†^	−0.020	0.287		Protein ^†^	−0.017	0.357
	Minerals	Potassium ^†^	0.003	0.880	Minerals	Potassium ^†^	−0.002	0.925
		Calcium ^†^	0.009	0.623		Calcium ^†^	0.006	0.728
		Phosphorus ^‡^	−0.033	0.079		Phosphorus ^‡^	−0.024	0.207
	Vitamins	Total vitamin A ^‡^	−0.006	0.731	Vitamins	Total vitamin A ^‡^	−0.002	0.894
		Retinol only ^‡^	−0.038	0.044		Retinol only ^‡^	−0.023	0.213
		Vitamin C ^‡^	−0.003	0.872		Vitamin C ^‡^	−0.013	0.502
	Water	Water ^‡^	−0.002	0.907	Water	Water ^‡^	0.004	0.825

^a^ Adjusted for age, BMI, physical activity and alcohol consumption. ^b^ Adjusted for age. ^†^ Parametric values were analysed by Pearson correlation. ^‡^ Non-parametric values were analysed by Spearman correlation. E.C.P., energy, carbohydrates, and protein.

**Table 10 ijerph-18-11658-t010:** Partial correlation coefficients between nutrient intake and bone mineral density in postmenopausal women.

BMD		Unadjusted	r	*p*		Adjusted	r	*p*
Total femur ^a^	E.C.P	Energy ^‡^	0.105	<0.001	E.C.P	Energy ^‡^	0.025	0.364
		Carbohydrate ^‡^	0.061	0.028		Carbohydrate ^‡^	0.037	0.182
		Protein ^‡^	0.154	<0.001		Protein ^‡^	0.027	0.334
	Minerals	Potassium ^‡^	0.172	<0.001	Minerals	Potassium ^‡^	0.065	0.020
		Calcium ^†^	0.154	<0.001		Calcium ^†^	0.064	0.021
		Phosphorus ^‡^	0.145	<0.001		Phosphorus ^‡^	0.041	0.144
	Vitamins	Total vitamin A ^‡^	0.144	<0.001	Vitamins	Total vitamin A ^‡^	0.038	0.173
		Retinol only ^‡^	0.148	<0.001		Retinol only ^‡^	0.055	0.050
		Vitamin C ^‡^	0.167	<0.001		Vitamin C ^‡^	0.053	0.058
	Water	Water ^‡^	0.208	<0.001	Water	Water ^‡^	0.073	0.009
Femoral neck ^a^	E.C.P	Energy ^‡^	0.116	<0.001	E.C.P	Energy ^‡^	0.034	0.221
		Carbohydrate ^‡^	0.063	0.024		Carbohydrate ^‡^	0.038	0.173
		Protein ^‡^	0.172	<0.001		Protein ^‡^	0.049	0.081
	Minerals	Potassium ^‡^	0.180	<0.001	Minerals	Potassium ^‡^	0.082	0.003
		Calcium ^†^	0.173	<0.001		Calcium ^†^	0.080	0.004
		Phosphorus ^‡^	0.162	<0.001		Phosphorus ^‡^	0.063	0.024
	Vitamins	Total vitamin A ^‡^	0.166	<0.001	Vitamins	Total vitamin A ^‡^	0.059	0.034
		Retinol only ^‡^	0.169	<0.001		Retinol only ^‡^	0.040	0.150
		Vitamin C ^‡^	0.174	<0.001		Vitamin C ^‡^	0.056	0.044
	Water	Water ^‡^	0.218	<0.001	Water	Water ^‡^	0.082	0.003
Lumbar spine ^b^	E.C.P	Energy ^‡^	0.073	0.009	E.C.P	Energy ^‡^	−0.008	0.770
		Carbohydrate ^‡^	0.002	0.946		Carbohydrate ^‡^	−0.027	0.325
		Protein ^‡^	0.143	<0.001		Protein ^‡^	0.024	0.381
	Minerals	Potassium ^‡^	0.148	<0.001	Minerals	Potassium ^‡^	0.046	0.095
		Calcium ^†^	0.174	<0.001		Calcium ^†^	0.092	0.001
		Phosphorus ^‡^	0.138	<0.001		Phosphorus ^‡^	0.039	0.165
	Vitamins	Total vitamin A ^‡^	0.134	<0.001	Vitamins	Total vitamin A ^‡^	0.026	0.355
		Retinol only ^‡^	0.176	<0.001		Retinol only ^‡^	0.019	0.505
		Vitamin C ^‡^	0.132	<0.001		Vitamin C ^‡^	0.019	0.484
	Water	Water ^‡^	0.190	<0.001	Water	Water ^‡^	0.066	0.017
Whole body ^b^	E.C.P	Energy ^‡^	0.071	0.011	E.C.P	Energy ^‡^	0.004	0.895
		Carbohydrate ^‡^	0.010	0.706		Carbohydrate ^‡^	−0.012	0.679
		Protein ^‡^	0.135	<0.001		Protein ^‡^	0.026	0.354
	Minerals	Potassium ^‡^	0.134	<0.001	Minerals	Potassium ^‡^	0.038	0.172
		Calcium ^†^	0.171	<0.001		Calcium ^†^	0.089	0.001
		Phosphorus ^‡^	0.131	<0.001		Phosphorus ^‡^	0.038	0.175
	Vitamins	Total vitamin A ^‡^	0.143	<0.001	Vitamins	Total vitamin A ^‡^	0.018	0.522
		Retinol only ^‡^	0.186	<0.001		Retinol only ^‡^	0.042	0.130
		Vitamin C ^‡^	0.120	<0.001		Vitamin C ^‡^	0.005	0.871
	Water	Water ^‡^	0.182	<0.001	Water	Water ^‡^	0.062	0.025

^a^ Adjusted for age, BMI, physical activity and alcohol consumption. ^b^ Adjusted for age. ^†^ Parametric values were analysed by Pearson correlation. ^‡^ Non-parametric values were analysed by Spearman correlation. E.C.P., energy, carbohydrates, and protein.

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
