# Peer review of "A Structural Equation Modelling Approach to Determine Factors of Bone Mineral Density in Korean Women"

_ijerph, 2021, doi:10.3390/ijerph182111658_

Round 1

Reviewer 1 Report

Dear Authors,

The submitted article is a significant and important source of information filling the gap in the knowledge of prevention and prophylaxis of osteoporosis. 

In my opinion, the article presents an very important health and social global problem. Osteoporotic fracture is a significant cause of morbidity and mortality.  Properly conducted research involving premenopausal (n=2863) and postmenopausal (n=1297) women aged over 30-75 years after using SEM led to the following conclusions:

1) There is a direct effect of minerals (potassium, calcium and phosphorus) on bone mineral density (BMD) (femur, femoral neck, lumbar spine and whole body) in pre- and postmenopausal women (P = 0.045 and P = 0.048, respectively).

2) Age and BMI showed a combined effect on BMD in pre- and postmenopausal women (P = 0.002, respectively).

The introduction provide sufficient background and include all relevant sources. The conclusions are supported by the results.

In my opinion, the study can be accepted after minor revision.

The correction should be made to the following lines:

95 - is vitamin A, retinol.  The term vitamin A includes all-trans retinol (also called retinol) and its derivatives: retinal, retinoic acid and retinyl esters (palmitate, propionate, acetate).

I would suggest using the terms ‘total vitamin A’ and ‘retinol only’

162, 183, 194, 209, 217, 220 and many other lines –p-value never really is exactly zero, should be p<0.001 instead of p=0.000.

212 – if deficiencies have been identified, the amount should be indicated.

 References appearing in the text should be adapted to the requirements of the editors and given in the form e.g. Lastname et al., 2017 or simply Lastname et al.

452-455 - should be deleted, it is a part of Microsoft Word template

461 - is vitamin, should be vitamins, in my opinion should be total vitamin A or retinol only

465 – is Feng et al. 2017, should be Feng et al., 2017 or simply Feng et al.

466 – is Rapuri et al 2003, should be Rapuri et al., 2003 or simply Rapuri et al.,

481 – note as to line 461

488- is vitamin A or retionol, in my opinion should be like at line 461

508 - note as to liness  465 and 466

517 - missing a dot point at the end of the sentence [16].

I believe that, with a little revision, this study can be published.

Author Response

The correction should be made to the following lines:

95 - is vitamin A, retinol.  The term vitamin A includes all-trans retinol (also called retinol) and its derivatives: retinal, retinoic acid and retinyl esters (palmitate, propionate, acetate).

I would suggest using the terms ‘total vitamin A’ and ‘retinol only’

→ Thank you for your comment.  

Through all the text, Vitamin A has been changed to “total vitamin A” and retinol has been changed to “retinol only”.

162, 183, 194, 209, 217, 220 and many other lines –p-value never really is exactly zero, should be p<0.001 instead of p=0.000.

→ Thank you. Now, P values are "P < 0.001".

212 – if deficiencies have been identified, the amount should be indicated.

 References appearing in the text should be adapted to the requirements of the editors and given in the form e.g. Lastname et al., 2017 or simply Lastname et al.

Thank you for your comment.

Now, references “[20], [21], [22]” have been included.

452-455 - should be deleted, it is a part of Microsoft Word template

Thank you for your comment. It has been deleted.

461 - is vitamin, should be vitamins, in my opinion should be total vitamin A or retinol only

 → Thank you for your comment.

Through all the text, Vitamin A has been changed to “total vitamin A” and retinol has been changed to “retinol only”.

465 – is Fung et al. 2017, should be Fung et al., 2017 or simply Fung et al.

Thank you for your comment. ‘Fung et al. 2017’ has been changed to “Fung et al., 2017”.

466 – is Rapuri et al 2003, should be Rapuri et al., 2003 or simply Rapuri et al.,

Thank you for your comment. ‘Rapuri et al 2003’ has been changed to “Rapuri et al., 2003”.

481 – note as to line 461

→ Thank you for your comment.

Through all the text, Vitamin A has been changed to “total vitamin A” and retinol has been changed to “retinol only”.

488- is vitamin A or retionol, in my opinion should be like at line 461

→ Thank you for your comment.

Through all the text, Vitamin A has been changed to “total vitamin A” and retinol has been changed to “retinol only”.

508 - note as to liness 465 and 466

Thank you for your comment.

‘Vannucci et al 2018’ has been changed to “Vannucci et al., 2018”.

517 - missing a dot point at the end of the sentence [16].

Thank you for your comment.

In line 588, It has been added a dot point at the end of the sentence: "~ through observed variables [17]."

Reviewer 2 Report

The present article was a very interesting read. Extensive English editing is, however, required. Alo, the following changes should be incorporated:

‘The country with highest percent of population aged over 65 years will have from 18 percent in 2000 to 38 percent in 2050’: Please be more specific, i.e., which country?

‘OECD’: Please explain each abbreviation at 1st mention.

‘…in 2050 in Asia as the aging population is 39 rapidly increasing in Asia’: Please reformulate (I suggest that the second use of the phrase in Asia in removed).

‘The fracture elevates mortality…’: I recommend reformulating as follows: Fractures elevate mortality rates…

Additional, extensive English editing is required. Several parts are completely incomprehensible. E.g., ‘Structural equation modelling (SEM) is a relatively powerful, multivariate technique by quantifying the associations and interactions among multiple factors.’ The secondary sentence ‘by…factors’ is not connected correctly with the rest of the text.

‘The SEM is beneficial to simultaneously examine all related pathways or associations between independent and /or dependent (e.g., mediators) variables’: Please replace independent and dependent with the terms exogenous and endogenous. The term dependent might be mistaken with the dependent variable under investigation by readers unfamiliar with the SEM technique.

‘Few studies exist assessing the associations of nutrient intake, physical activity, age, 56 body mass index (BMI) with BMD using SEM in women.’ In addition to the above statement, the factors you investigate are well-established determinants of BMD. Considering the above, what does the present study add?

Please describe the KNHANES studies in more detail: i.e., participant selection (random? stratified? etc), population of interest (adults? race? population-based design? etc), study design (prospective or retrospective? cohort?), etc.

Disease status was considered an exclusion criterion. Is there any residual confounding from less prevalent diseases, not accounted in the original design of the study? What about other (non-disease) parameters.

Exposure ascertainment: How were (1) income level (no description was provided) (2) education level (which are the educational landmarks of the investigated population?) (3) occupation status (‘yes – no’ is not an adequate description, does ‘no’ include both unemployed and pensioners? were recently unemployed adults classified as ‘no’? etc) (4) mean daily sleep time (was a direct question addressed?, was any specific period considered in this question, i.e., one month prior to examination?, etc) (5) smoking status (definitions and rationale, I have never encountered the criterion of 100 cigarettes in a lifetime before) and (6) alcohol intake (definitions and rationale) determined?

Please describe the characteristics of those with missing data (missing data analysis is important). Have you considered imputation?

Regarding dietary assessments, were findings from the ‘pass 24-hour dietary recall questionnaire for nutrient intake assessment’ considered representative of the general dietary habits of an individual?

‘3.2. Nutrient intakes provide a detailed descriptive approach for nutrient intake. Were these variables described in detail purposeful? What I mean is that the aim of the study was completely irrelevant to the recommended nutrient intakes (is this section really necessary?). In general, regarding 3.1 – 3.3, participant characteristics should be summarized in a Table (as done) and not described in detail in the text.

‘Authors should discuss the results and how they can be interpreted from the perspective of previous studies and of the working hypotheses. The findings and their implications should be discussed in the broadest context possible. Future research directions may also be highlighted.’ Please explain the introductory paragraph in the section Discussion.

‘So far, no study exists to examine determinants of 514 BMD with a SEM approach.’ This statement comes in contradiction with the above-mentioned statement that ‘Few studies exist assessing the associations of nutrient intake, physical activity, age, 56 body mass index (BMI) with BMD using SEM in women.’

Please provide the limitations of your study in more detail (non-response bias?, measurement or information bias?, missing data, residual confounding?, etc)

Author Response

The present article was a very interesting read. Extensive English editing is, however, required. Alo, the following changes should be incorporated:

‘The country with highest percent of population aged over 65 years will have from 18 percent in 2000 to 38 percent in 2050’: Please be more specific, i.e., which country?

→ Thank you for your comment. The sentence has been revised with a new reference [2] in lines 32-36.

“The number of persons aged over 60 years was 962 million in 2017 worldwide. This number is expected to double with 2.1 billion in 2050. In 2050, the persons aged over 60 years will account for 35 percent of the population in Europe, 28 percent in North-ern America, 25 percent in Latin [1,2].”

[2] Nations, U. World Population Ageing 2017 Highlights. Affairs, D.o.E.a.S., Ed. United Nations: New York, 2017.

‘OECD’: Please explain each abbreviation at 1st mention.

Thank you for your comment. It has been addressed as follows:

Organization for Economic Cooperation and Development (OECD)”

‘…in 2050 in Asia as the aging population is 39 rapidly increasing in Asia’: Please reformulate (I suggest that the second use of the phrase in Asia in removed).

→ Thank you. Second use of the phrase “in Asia” has been deleted.

‘The fracture elevates mortality…’: I recommend reformulating as follows: Fractures elevate mortality rates…

Thank you. It has been changed to in lines 46-48.

In lines 46-48

 “Fractures elevate mortality rates [8] and imposes an enormous burden on individuals (e.g., early retirement), society (e.g., work impact) and the health care system (e.g., hospitalizations, medication, rehabilitation and so on) [9-11].”

Additional, extensive English editing is required. Several parts are completely incomprehensible. E.g., ‘Structural equation modelling (SEM) is a relatively powerful, multivariate technique by quantifying the associations and interactions among multiple factors.’ The secondary sentence ‘by…factors’ is not connected correctly with the rest of the text.

Thank you for your comment. It has been changed to in lines 57-58.

In lines 57-58

 “Structural equation modelling (SEM) is a relatively powerful, multivariate technique that quantifies the associations and interactions between multiple variables.”

‘The SEM is beneficial to simultaneously examine all related pathways or associations between independent and /or dependent (e.g., mediators) variables’: Please replace independent and dependent with the terms exogenous and endogenous. The term dependent might be mistaken with the dependent variable under investigation by readers unfamiliar with the SEM technique.

Thank you for your comment. It has been changed to in lines 60-62.

In lines 60-62

“The SEM is beneficial to simultaneously examine all related pathways or associations between exogenous and /or endogenous (e.g., mediators) variables [17].”

‘Few studies exist assessing the associations of nutrient intake, physical activity, age, body mass index (BMI) with BMD using SEM in women.’ In addition to the above statement, the factors you investigate are well-established determinants of BMD. Considering the above, what does the present study add?

Thank you for your comment. The statement we wrote made readers to be confused. Therefore, the sentence has been changed in lines 63-64.

In lines 63-64

 “No studies exist assessing the associations of nutrient intake, physical activity, age, body mass index (BMI) with BMD using SEM in Korean women.”

Please describe the KNHANES studies in more detail: i.e., participant selection (random? stratified? etc), population of interest (adults? race? population-based design? etc), study design (prospective or retrospective? cohort?), etc.

Thank you. It has been addressed in lines 71-76.

In lines 71-76

The KNHANES is a national, cross-sectional health examination and survey in Korean general population [18]. This survey protocol used a stratified, multistage clustered probability sampling method for a representative sample of the noninstitutionalized Korean population. Systematic sampling is annually conducted with a new and different sample of about 10,000 persons aged 1 year and over.”

Disease status was considered an exclusion criterion. Is there any residual confounding from less prevalent diseases, not accounted in the original design of the study? What about other (non-disease) parameters. 

Thank you for your comment.

Diseases involved in bone metabolism were used as excluded variables as mentioned in the text. Other parameters were included for this SEM analysis. Therefore, in this study, only the variables [physical activity (high intense physical activity, moderate intense physical activity and regular walking), age, BMI, E.C.P. (energy, carbohydrate and protein), minerals (potassium, calcium and phosphorus) and vitamins (vitamin A, retinol and vitamin C)] for the total or mediating effects with structural equations are shown as pathways.

It has been addressed in limitation of this study in lines 602-603 and 605-609.

In lines 602-603

“Bias for non-response or measurement or information was not ruled out because this study was based on cross-sectional study. “

In lines 605-609

“Residual confounding could be also one of limitations. However, a SEM analysis considers the measurement error of the measured variable and the perturbation error of the latent variable, so that the reflected value can be analyzed. In this aspect, a SEM has the advantage of analyzing values considering errors compared with a logistic regression analysis.”

Exposure ascertainment: How were (1) income level (no description was provided) (2) education level (which are the educational landmarks of the investigated population?) (3) occupation status (‘yes – no’ is not an adequate description, does ‘no’ include both unemployed and pensioners? were recently unemployed adults classified as ‘no’? etc) (4) mean daily sleep time (was a direct question addressed?, was any specific period considered in this question, i.e., one month prior to examination?, etc) (5) smoking status (definitions and rationale, I have never encountered the criterion of 100 cigarettes in a lifetime before) and (6) alcohol intake (definitions and rationale) determined?

Thank you. This study was conducted based on the questionnaire of Korea National Health and Nutrition Examination Survey (KNHANES). It has been addressed in lines 71-76 and lines 78-103.

In lines 71-76

The KNHANES is a national, cross-sectional health examination and survey in Korean general population [18]. This survey protocol used a stratified, multistage clustered probability sampling method for a representative sample of the noninstitutionalized Korean population. Systematic sampling is annually conducted with a new and different sample of about 10,000 persons aged 1 year and over.”

In lines 78-103

“The health interview consists of components including housing characteristics, medical conditions, socioeconomic status (income level, education level, occupation status), health care utilization, activity limitation, quality of life and injury, smoking, alcohol use, physical activity, mental health (mean daily sleep time), oral health, weight control, safety, reproductive health for women. Income level was categorized as high, middle-high, middle-low, or low. Education level was categorized as less than elementary school, middle school, high school, or more than university. Occupation status was divided into “yes” for the employed and “no” for the unemployed and economically inactive population, which was based on the question: “Have you worked for more than an hour for income or for more than 18 hours as an unpaid family worker in the past week? You need to answer “yes” if you are temporarily on leave from your work”. The question on the mean daily sleep time was “How much hours per day do you usually sleep? You need to write the mean daily sleep time”. The question on smoking status was categorized into non-smoker, former smoker and current smoker during the lifetime. Subjects were asked if they never smoke or they smoked in the past (yes) or they currently smoke (yes) during the lifetime. If subjects answered “yes”, they chose the one of the following answers: less than 5 packs (100 cigarettes) or more than 5 packs (100 cigarettes). The question on alcohol intake was categorized into non-drinker (no) and drinker (yes) during the lifetime based on “Have you ever had more than one drink in your lifetime?”. The health examination consists of components including body measurements, blood pressure, laboratory test (blood and urine), dental measurement, vision, retinal photo and visual field, audiometry, spirometry, balance, bone density and body composition, chest, knee and hip-joint X-ray. The nutrition survey consists of components including dietary behavior, dietary supplement use, food security, food frequency, food and dietary intake [18]. Detailed explanation on the KNHANES is available at http://knhanes.kdca.go.kr.”

Please describe the characteristics of those with missing data (missing data analysis is important). Have you considered imputation?

Thank you for your comment. It has been addressed in line 182-184.

In lines 182-184

Before performing SEM procedures, missing values were treated through data preprocessing process. The processing of missing data is typically performed by removing or replacing, but in this study, all analyzes were performed by removing missing values.”

Regarding dietary assessments, were findings from the ‘pass 24-hour dietary recall questionnaire for nutrient intake assessment’ considered representative of the general dietary habits of an individual?

Thank you for your comment. It has been addressed in limitation in lines 603-604.

In lines 603-604

“The pass 24-hour dietary recall for nutrient intake assessment might not be representative of the general dietary habits of an individual.”

‘3.2. Nutrient intakes provide a detailed descriptive approach for nutrient intake. Were these variables described in detail purposeful? What I mean is that the aim of the study was completely irrelevant to the recommended nutrient intakes (is this section really necessary?).

Thank you for your comment.

We believe that the detailed description on nutrient intake could be helpful for readers (including Korean readers) to understand the nutrient intake status in Korean women. One of reviewers also asked us to cite references for the 2020 dietary reference intakes for Koreans (KDRIs). For this reason, we would like to leave the detailed information in section 3.2 Nutrient intakes.

In general, regarding 3.1 – 3.3, participant characteristics should be summarized in a Table (as done) and not described in detail in the text.

Thank you for your comment. Section 3.1 and 3.3 has been merged. The description has been summarized by deleting several sentences. Please see in lines 205-220.

‘Authors should discuss the results and how they can be interpreted from the perspective of previous studies and of the working hypotheses. The findings and their implications should be discussed in the broadest context possible. Future research directions may also be highlighted.’ Please explain the introductory paragraph in the section Discussion.

Thank you. It has been addressed in lines 564-583 and 611-612.

In lines 564-583

“The present study of models indicated a high goodness of fit with a TLI of 0.949, a CFI of 0.967 and a RMSEA of 0.059 for premenopausal and postmenopausal women, and with a TLI of 0.924, a CFI of 0.945 and a RMSEA of 0.079 for postmenopausal women. This stable and appropriate model fit strongly supported our research hypotheses that nutrient intakes, physical activities, age and BMI would exert total, direct and mediating effects on BMD in Korean premenopausal and postmenopausal women aged between 30 and 75 years. In summary of our findings, a direct effect of minerals (potassium, calcium and phosphorus) on BMD (total femur, femoral neck, lumbar spine and whole body) was observed in premenopausal and postmenopausal women. A direct effect and a total effect of age and BMI on BMD was observed in premenopausal and postmenopausal women. In premenopausal women, vitamin intake (total vitamin A, retinol only and vitamin C), water intake and E.C.P (energy, carbohydrate and protein) intake showed a direct effect on BMD. E.C.P. intake and vitamins intake exerted a mediating effect on the association between age and BMD. E.C.P. intake showed a mediating effect on the association between BMI and BMD. Our findings supported the results of previous studies [25, 38, 48] which showed the favorable influence of minerals on bone health. Moreover, our findings indicate age, BMI and mineral intakes, which are major factors, can interact with each other, leading to influence on bone health. The findings of this study can aid the development of nutrition education and lifestyle modification strategies for the prevention from osteoporosis and fractures in aging women.”

In lines 611-612

“Longitudinal and intervention studies including the variables covered in this study should be conducted in the future.”

‘So far, no study exists to examine determinants of 514 BMD with a SEM approach.’ This statement comes in contradiction with the above-mentioned statement that ‘Few studies exist assessing the associations of nutrient intake, physical activity, age, 56 body mass index (BMI) with BMD using SEM in women.’

Thank you. It has been addressed in lines 585-586.

In lines 585-586

“So far, no study exists to examine determinants of BMD with a SEM approach in Korean women.”

Please provide the limitations of your study in more detail (non-response bias?, measurement or information bias?, missing data, residual confounding?, etc)

Thank you. It has been addressed in lines 602-609.

In lines 602-609

“Bias for non-response or measurement or information was not ruled out because this study was based on cross-sectional study. The pass 24-hour dietary recall for nutrient intake assessment might not be representative of the general dietary habits of an individual. Residual confounding could be also one of limitations. However, a SEM analysis considers the measurement error of the measured variable and the perturbation error of the latent variable, so that the reflected value can be analyzed. In this aspect, a SEM has the advantage of analyzing values considering errors compared with a logistic regression analysis.”

Round 2

Reviewer 2 Report

Thank you for addressing my considerations.